# The Relationship between 9/11 Exposure, Systemic Autoimmune Disease, and Post-Traumatic Stress Disorder: A Mediational Analysis

**DOI:** 10.3390/ijerph19116514

**Published:** 2022-05-27

**Authors:** Jennifer Brite, Sara A. Miller-Archie, James Cone

**Affiliations:** 1New York City Department of Health and Mental Hygiene, New York, NY 11101, USA; smille12@health.nyc.gov (S.A.M.-A.); jcone@health.nyc.gov (J.C.); 2School of Health Sciences and Professional Programs, York College, The City University of New York (CUNY), New York, NY 11451, USA

**Keywords:** systemic autoimmune disease, mediation analysis, post-traumatic stress disorder

## Abstract

The relationship between 9/11 exposure, systemic autoimmune disease (SAD) and mental health remains poorly understood. This report builds on a prior analysis of World Trade Center Health Registry data to determine whether 9/11 exposure is associated with higher risk of SAD, and if so, whether post-traumatic stress disorder (PTSD) is a mediating factor and whether the association varies by responder/community member status. The final analytic sample comprised 41,656 enrollees with 123 cases of SAD diagnosed post 9/11 through November 2017. SAD diagnosis was ascertained from survey responses and confirmed by medical record review or physician survey. Logistic regression models were constructed to determine the relationship between 9/11 exposure and PTSD and SAD. Causal mediation analysis was used to determine the mediational effect of PTSD. Each analysis was stratified by 9/11 responder/community member status. Rheumatoid arthritis (*n* = 75) was the most frequent SAD, followed by Sjögren’s syndrome (*n* = 23), systemic lupus erythematosus (*n* = 20), myositis (*n* = 9), mixed connective tissue disease (*n* = 7), and scleroderma (*n* = 4). In the pooled cohort, those with 9/11-related PTSD had 1.85 times the odds (95% CI: 1.21–2.78) of SAD. Among responders, those with dust cloud exposure had almost twice the odds of SAD, while among community members, those with 9/11-related PTSD had 2.5 times the odds of SAD (95% CI: 1.39, 4.39). PTSD was not a significant mediator. Although emerging evidence suggests 9/11 exposure may be associated with SAD, more research is needed, particularly using pooled data sources from other 9/11-exposed cohorts, to fully characterize this relationship.

## 1. Introduction

Autoimmune disorders are often debilitating, require life-long care, and generally have no definitive cure. There are more than 80 known autoimmune disorders, and in each the immune system mistakenly targets healthy cells or organs. Autoimmune disorders are more prevalent in women, particularly minorities, and worldwide incidence may be increasing [1]. Systemic autoimmune diseases (SAD) are those in which an abnormal immune response is found in almost any type of cell in the body and is not localized.

A growing body of literature suggests rates of SAD may be higher among those with higher versus lower 9/11 exposure, particularly responders [2,3,4]. However, results have been conflicting [5] and the mechanisms underlying this association remain unclear.

Although approximately 50% of autoimmune cases are of unknown origin [6], two broad classes of September 11-related exposures have been hypothesized to be biologically plausible etiologies of subsequent autoimmune disease. Environmental exposures represent the first. The collapse of the towers caused a large release of a dust cloud and other debris that covered much of lower Manhattan for several months after the event throughout recovery and clean-up [7]. Several compounds that have been previously associated with autoimmune disease [8] have been found to be present in the dust cloud, including organic hydrocarbon solvents [9], fine particulate matter [10], and asbestos [11]. However, the exact etiology and pathogenesis between these environmental exposures and SAD remains unknown, and genetic susceptibility may play a role [12].

Another hypothesized pathway works through diminished mental health following a traumatic event. SAD manifests as an abnormal immune reaction in specific organs or bodily systems, and may be influenced by psychological reactions to life stressors [13]. Post-traumatic stress disorder (PTSD) has been shown to be strongly associated with SAD in the World Trade Center Health Registry cohort (WTCHR) [2]. In addition, new-onset SAD has been found to be more common among those exposed to high levels of trauma, in both veteran and civilian populations [14,15,16,17,18]. For example, a recent study using the Millennium Cohort Study data found the hazard of developing an autoimmune disease confirmed by medical records was 58% higher for those with a history of PTSD (HR = 1.58, 95% CI: 1.25, 2.01) compared to those with no history of PTSD [19].

However, the contribution of one or more of these causal pathways to the development of post-trauma SAD has not been thoroughly described. The present analysis examines the relationship between 9/11 exposure and subsequent autoimmune disease by determining the mediational effect of PTSD using WTCHR data. No study, to our knowledge, has formally tested the mediational relationships between exposure to 9/11, PTSD, and subsequent SAD.

This study aims 1. to characterize the association between 9/11 exposure and SAD using additional data not available in the original report on this topic in the WTCHR cohort; 2. to determine whether PTSD mediates the association between World Trade Center-disaster exposure and SAD; and 3. to determine whether the association differs by responder versus community member status.

## 2. Materials & Methods

### 2.1. WTC Health Registry

The WTCHR is a longitudinal, prospective cohort of 71,424 enrollees who were exposed to the 9/11 (11 September 2001) World Trade Center attacks in New York City, or who were involved in the subsequent recovery and clean-up effort, which lasted until July 2002. The WTCHR includes people who were part of the rescue, recovery, and clean-up response (responders = 43%) or worked, resided, attended school, or were in transit in lower Manhattan on the morning of the attacks (community members = 57%). Enrollment occurred between 2003 and 2004, after which four subsequent waves of data collection took place. A more detailed description of the Registry’s recruitment methods has been published elsewhere [20]. The study was approved by the Institutional Review Board (IRB) of the New York City Department of Health and Mental Hygiene and the IRB for the Centers for Disease Control and Prevention.

### 2.2. Autoimmune Disease Sub-Study

An autoimmune study was conducted among adult WTCHR enrollees who self-reported being diagnosed with rheumatoid arthritis or another autoimmune disorder post-9/11 on the wave 3 survey conducted between 2011 and 2012. A detailed description of the study methods has been previously published [2]. In brief, enrollees were sent an in-depth autoimmune survey in May 2014. Those who completed the survey (*n* = 2042) were then sent a letter with a consent form authorizing the release of medical records to the WTCHR. Each case was confirmed either via a survey initially sent to the enrollee’s physician in May 2015 or a medical record review conducted by WTCHR staff. The response rate for the enrollee survey was 73%, of whom 51% consented to medical record review.

A previously published report presented data based on the medical records and physician surveys obtained as of November 2017 [2]. The current analysis includes all medical records and physician surveys received by the Registry and includes an additional approximately 200 patient data files and five cases.

### 2.3. Analytic Sample

Only enrollees who responded to wave 3 and were 18 or older on 9/11 were included in the study (*n* = 42,394). Those who responded affirmatively to SAD in the wave 3 survey but were not confirmed by medical record review or had a diagnosis before 2002 were excluded, resulting in an analytic sample of 41,656. For the mediation analysis, confirmed SAD cases with diagnosis dates before wave 1 were excluded.

### 2.4. Outcome

The outcome of interest was post-9/11 SAD, including rheumatoid arthritis (RA), systemic lupus erythematosus (SLE), scleroderma, mixed connective tissue disease (MCTD), polymyositis, dermatomyositis, or Sjögren’s Syndrome (SS). All cases were confirmed either by medical chart review or physician survey.

### 2.5. Exposure

Exposure to the 11 September 2001, disaster was assessed in two ways. First, intensity of dust cloud exposure was determined using responses to the waves 1 and 2 surveys. Those who reported being caught in the dust cloud and experiencing the inability to see a few feet ahead, trouble walking or navigating due to dust cloud thickness, taking shelter from the dust cloud, being covered in dust and debris, or the inability to hear anything in the dust cloud were classified as having had intense dust cloud exposure. All others were classified as low/no dust cloud exposure.

In the second measure of 9/11 exposure, “composite dust score”, indices were developed by a modified Delphi method, in which a panel of industrial hygienists and occupational health epidemiologists evaluated a variety of enrollee responses to questions related to dust exposure that were collected on waves 1 and 2. They included questions about whether an enrollee smelled smoke on 9/11, whether they were caught in the dust cloud, and if so when, and, for residents, whether they cleaned dust in their home. A scale of 1 to 10 was assigned to each exposure item. Respondents scores were then dichotomized into high (above 31.85 or 75th percentile) or low.

September 11-related PTSD was the proposed mediating variable and was assessed at Registry enrollment using a 9/11-specific modified version of the PTSD Checklist (PCL). To maintain consistency with a previous analysis of this cohort, a score ≥ 44 was considered probable PTSD; all scores < 44 were classified as no PTSD [2]. Participants with missing responses to the PCL were classified in the appropriate category if their PTSD status could be identified based on completed items. All others were categorized as missing PTSD.

### 2.6. Covariates

Covariates included age at 9/11, race/ethnicity, gender, marital status, household income, education, and smoking status. All were self-reported at enrollment. Race/ethnicity was mutually exclusive and classified as White, Black or African American, Hispanic or Latino, and Asian/Multiracial/Other. Marital status was classified as married or living with partner, divorced, separated, or widowed, or never married. Household income was classified as <$35,000, $35,000 to <$50,000, $50,000 to <$75,000, $75,000 to <$100,000, $100,000 to <$150,000, or $150,000 or more. Educational attainment was categorized as no high school diploma, high school diploma, some college, or at least a Bachelor’s degree. Smoking status was classified as ever having smoked or not. All covariates were chosen a priori based on a review of scientific literature.

### 2.7. Statistical Analysis

Bivariate analyses were conducted either using the χ^2^ test or Fisher’s exact test when expected cell counts were less than 5 to compare the distribution of 9/11 exposure and sociodemographic factors between those who had SAD and those who did not. For aim 1, logistic regression models were constructed to assess the odds of SAD by 9/11 exposure status. For aim 2, a mediational analysis was conducted to determine whether PTSD mediated the association between 9/11 exposure and SAD. The mediational methods used here have been fully described by Imai et al. [21]. Briefly, this method calculates average mediation and direct effect sizes by simulating the predicted potential of developing SAD and PTSD and then decomposing those effects into direct, indirect, and total effect sizes. In addition, percentage mediated was calculated as: indirect effect/(direct effect + indirect effect) × 100.

All models were fully adjusted with all covariates. All analyses were completed in SAS 9.4, Cary, North Carolina, USA, and RStudio version 1.3.959 using the R mediation package.

## 3. Results

The final analytic sample included 41,656 participants, of whom 123 had SAD. Those with SAD were slightly older (median age at 9/11: 44.2 vs. 42.4 years) and more likely to be non-White and female compared to those without SAD. No statistically significant differences were found between SAD and non-SAD participants for marital status, household income, educational attainment, or smoking status (Table 1).

Among those with SAD, RA was the most common diagnosis (*n* = 75), followed by SS and SLE (23 and 20 confirmed cases respectively). There were fewer than 10 cases each of myositis, MCTD, and scleroderma.

In multivariable analysis, in the total analytic sample, only 9/11-related PTSD was statistically significantly associated with SAD (OR: 1.85 [95% CI: 1.21, 2.78]). The odds ratios for both the dust intensity and composite dust score were elevated, though neither reached statistical significance. Among responders, those with high dust intensity exposure had 1.99 (95% CI: 1.10, 3.45) times the odds of SAD compared to those with low or no exposure. Among community members, those with probable 9/11-related PTSD had 2.50 (95% CI: 1.39, 4.39) times the odds of SAD compared to those with no 9/11-related PTSD (Table 2).

Although several of the 9/11 exposure-SAD associations were not statistically significant, a mediation analysis was conducted because mediation may be present even when main effects are non-significant, particularly when a study outcome is rare [22]. In the mediation analysis, 9/11-related PTSD did not mediate any association between 9/11 exposure measure and SAD in the full cohort, or when stratified by responder vs. community member status (data not shown).

## 4. Conclusions

This study found 9/11-related PTSD was statistically significantly associated with systemic autoimmune disease (SAD) among community members. High dust intensity exposure was associated with an increased risk of SAD only among responders. These findings are similar to those found in a previous examination of Registry participants conducted before all self-reported cases prior to wave 3 were fully confirmed by medical chart review or physicians survey [2]. Similar to the current analysis, that study also examined a composite dust score for community members that was not based on the Delphi method. The current study uses a Delphi exposure method that is similar for both responders and community members. Both studies found the dust composite score was not associated with SAD in either stratified or pooled analyses. These results are also similar to those of two studies that enrolled only Fire Department of New York (FDNY) workers, mostly male firefighters, all of whom were exposed to 9/11. Those studies found increased duration of rescue and recovery work was associated with increased odds of SAD [3,4]. However, a study of all potential autoimmune disease cases, not limited to systemic autoimmune diseases, among general responders that included a larger number of cases (*n* = 734, including definitive and probable cases), but few FDNY workers, found no elevated risk of SAD with increasing 9/11 exposure. PTSD was adjusted for in that analysis but was not statistically significant [5].

This study utilized several measures of 9/11 exposure, and results were not consistent across measures. Although the only 9/11 exposure variable that reached statistical significance was dust intensity in responders, it is important to note all point estimates were elevated and the confidence intervals are relatively wide. The comparatively small number of cases reduced the power of this study to detect positive associations between exposure and outcomes. Specifically, the dust cloud has been found to be composed of chemical compounds associated with SAD, including crystalline silica [11,23,24,25,26,27,28] organic hydrocarbon solvents [9,29,30], fine particulate matter [10,31,32], and asbestos [11,33]. The community group consists of both residents who eventually returned to lower Manhattan, and also passersby and office workers who never returned to the disaster site or returned when cleanup had commenced. In contrast, the majority of responders spent considerable time working in close proximity to “the pile.” It is possible that the longer average duration of dust exposure among responders may explain the stronger association between dust cloud exposure and SAD among responders compared to community members.

In addition to environmental exposures, the traumatic events of 9/11 have been linked to several detrimental health effects, particularly PTSD [34,35,36]. PTSD has been shown to cause systemic inflammation resulting in adverse health outcomes [14,15,16,17,18]. However, there have been few studies examining the potential mechanisms of PTSD and autoimmune disease in particular. One study found that war veterans with PTSD exhibited a different phenotype of Regulatory T cells (Tregs), which may prevent the development of autoimmune diseases, compared with healthy controls [37]. Inflammation may also play a role. For example, in another study, Vietnam war veterans suffering from comorbid PTSD were more likely to have hyperreactive immune responses on standardized delayed cutaneous hypersensitivity tests, clinically higher immunogolobulin-M levels, and clinically lower dehydroepiandrosterone levels. As was found in previous studies on this topic, 9/11-related PTSD was associated with SAD in both responders and community members. However, this association was significant only among community members. This suggests mental health may be a more important causal pathway for development of SAD in community members, who were more likely to be non-White, female, and to suffer from PTSD. More research is needed to determine why causal pathways may differ in responders and community members.

Previous work has examined World Trade Center exposures, such as dust cloud exposure, and 9/11-related PTSD as independent predictors. The present analysis adds to the existing literature by examining whether PTSD mediates the association between 9/11 exposure and SAD. In other words, we attempted to determine whether being exposed to 9/11 increased the risk of SAD net of subsequent mental health effects or whether 9/11 PTSD was driving any observed association. PTSD was not a significant mediator in the relationship between any 9/11 exposure and SAD. These results suggest dust cloud exposure may have played a role independent of PTSD in responders (the only group for whom dust cloud exposure was statistically significant). It is unclear whether other stress-related mental health conditions may have similar relationships with 9/11 exposure and SAD.

This study had the following limitations: First, our sample only included respondents to the wave 3 survey. Enrollees who were lost to follow-up or who died between waves 1 and 3 were therefore excluded. Second, approximately one half of the enrollees who responded to the autoimmune in-depth survey did not provide consent for either the physician survey or medical record review. A previous report found those who did provide consent were more likely to be male and a responder, although there were no significant differences by dust exposure status or PTSD [2]. Finally, this analysis included a relatively small number of cases, and our measures of 9/11 exposures were constructed using self-report data.

This study also had several strengths. All cases of SAD were confirmed by physician survey or medical record review. This is particularly important as SAD is overreported on self-reported surveys [38,39,40]. Additionally, this study had a large sample size, high response rates to the autoimmune in-depth study, and a prospective design.

The relationship between 9/11 exposures, SAD, and PTSD will require further study. Given the relatively small number of cases, using data from additional sources, such as FDNY and the General Responder Cohort (which consists of workers and volunteers that were part of the rescue/recovery effort), may be useful. Combining data from multiple sources will allow for increased power, more stable effect estimates, analysis of sub-groups, and better data triangulation.

## Figures and Tables

**Table 1 ijerph-19-06514-t001:** Analytic sample characteristics of the World Trade Center Health Registry enrollees stratified by systemic autoimmune disease status.

	No Autoimmune Disease (*N* = 41,533)	Autoimmune Disease (*N* = 123)	*p*-Value
9/11-related PTSD: yes	6273 (15.1%)	35 (28.5%)	<0.001 *
High Dust Intensity (from survey)	10,540 (25.4%)	41 (33.3%)	0.0693
High Dust Composite Score (from Delphi method)	8562 (20.6%)	33 (26.8%)	0.112
Responder/Community Member Status			
Community Member	22,013 (53.0%)	56 (45.5%)	0.117
Responder	19,520 (47.0%)	67 (54.5%)	
Age at 9/11 Mean (SD)	42.4 (11.3)	44.2 (9.68)	0.0328 *
Race/ethnicity			
White (non-Hispanic)	28,722 (69.2%)	79 (64.2%)	0.0571
Black or African American (non-Hispanic)	4234 (10.2%)	18 (14.6%)	
Hispanic or Latino (any race)	4805 (11.6%)	20 (16.3%)	
Asian/Multiracial/Other	3772 (9.1%)	6 (4.9%)	
Male	25,477 (61.3%)	42 (34.1%)	<0.001 *
Marital status			
Married or living with partner	27,307 (65.8%)	80 (65.0%)	0.145
Divorced or separated or widowed	5555 (13.4%)	23 (18.7%)	
Never married	8295 (20.0%)	19 (15.4%)	
Household income			
Less than $35,000	5839 (14.1%)	14 (11.4%)	0.104
$35,000 to less than $50,000	4822 (11.6%)	16 (13.0%)	
$50,000 to less than $75,000	8240 (19.8%)	28 (22.8%)	
$75,000 to less than $100,000	7429 (17.9%)	27 (22.0%)	
$100,000 to less than $150,000	6531 (15.7%)	25 (20.3%)	
$150,000 or more	4652 (11.2%)	6 (4.9%)	
Educational attainment			
No high school diploma	1514 (3.6%)	5 (4.1%)	0.098
High school diploma	7502 (18.1%)	29 (23.6%)	
Some college	10,128 (24.4%)	38 (30.9%)	
At least a Bachelor’s	22,052 (53.1%)	50 (40.7%)	
Ever smoke: yes	23,523 (56.6%)	62.0 (50.4%)	0.378

Some counts may not total study population due to missing data. N and percentage unless otherwise specified. * Statistically significant at 0.05 level.

**Table 2 ijerph-19-06514-t002:** Regression results (OR) of the association between 9/11 exposure and PTSD and systemic autoimmune disease ^a^.

9/11 Exposure ^b^	Total Cohort OR (95% CI)	Responder OR (95% CI)	Community Member OR (95% CI)
Dust intensity (high vs. low)	1.42 (0.96, 2.08)	1.99 (1.10, 3.45) *	1.44 (0.82, 2.49)
Dust composite score (high vs. low)	1.35 (0.89, 2.00)	1.65 (0.83, 3.02)	1.60 (0.91, 2.74)
PTSD (yes vs. no)	1.85 (1.21, 2.78) *	1.52 (0.77, 2.79)	2.50 (1.39, 4.39) *

^a.^ adjusted for age, race, gender, marital status, income, education, and smoking history ^b.^ Individual models were created for each 9/11 exposure * statistically significant.

## Data Availability

The data presented in this study are available on request from the World Trade Center Health Registry.

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
