# Peer review of "The Relationship between 9/11 Exposure, Systemic Autoimmune Disease, and Post-Traumatic Stress Disorder: A Mediational Analysis"

_ijerph, 2022, doi:10.3390/ijerph19116514_

Round 1

Reviewer 1 Report

The authors have addressed the concerns of my previous review.

Author Response

Thank you for your comments and improvements.

Reviewer 2 Report

The type of the manuscript is a Brief report. With that, the manuscript is about 9 pages. I am not convinced that such a manuscript can be considered as a Brief report.

The authors have found no new or additional data. The sample size is too small (35 out of 6308 subjects) to make any justified conclusions.

Author Response

Thank you for your comments and improvements. 

Point 1. 

The type of the manuscript is a Brief report. With that, the manuscript is about 9 pages. I am not convinced that such a manuscript can be considered as a Brief report.

We are unaware of the page and word limitations for Brief reports to this journal. We are happy to amend the manuscript if given this information. 

Point 2. The authors have found no new or additional data.

A previous manuscript using Registry data was published using interim data due to scientific interest in this topic, as well as demand by WTC survivors. The present manuscript represents the full dataset and includes an additional 200 case files and 5 cases. We have updated the manuscript in the first paragraph of page 3 to better clarify this issue. 

Point 3. The sample size is too small (35 out of 6308 subjects) to make any justified conclusions.

We agree. The discussion section outlines the issues around lack of power inherent in a study with this relatively small number cases. We have also added this to the limitations section. 

This manuscript is a resubmission of an earlier submission. The following is a list of the peer review reports and author responses from that submission.

Round 1

Reviewer 1 Report

The introduction is not clear and should be rewritten with the comprehensive analysis of a subject and appropriate references

 For the comparison between groups, the authors used the x-test (X-test should not be used to compare samples of 120 and 28,000 people),

The discussion is also written in some scattered pieces of interpretations.

The main findings of the study were previously published.

Reviewer 2 Report

The authors found that a total of 41,656 participants, 123 subjects out of whom had systemic autoimmune disease (SAD), and 35 subjects (28.5%) were related to 9/11-related PTSD. In contrast, in the non-autoimmune disease category (n=41533), there were 6273 subjects (15.1%) associated with 9/11-related PTSD. Particularly, one might assume that the 6273 subjects that had prior 9/11-related PTSD would be highly associated with SAD.

Although authors have specifically stated that the SAD (n=35 subjects) was acquired post-9/11, in view of the total subjects for 9/11-related PTSD (n=6308; n=6273 non-autoimmune + n=35 SAD), it is only 0.55% of subjects (35/6308 * 100%) with 9/11-related PTSD exposure developed SAD. Reviewer is not convinced that 9/11 exposure is associated with higher risk of SAD.